# Meeting Challenges of Pediatric Drug Delivery: The Potential of Orally Fast Disintegrating Tablets for Infants and Children

**DOI:** 10.3390/pharmaceutics15041033

**Published:** 2023-03-23

**Authors:** Klervi Golhen, Michael Buettcher, Jonas Kost, Jörg Huwyler, Marc Pfister

**Affiliations:** 1Pediatric Pharmacology and Pharmacometrics, University Children’s Hospital Basel (UKBB), University of Basel, 4056 Basel, Switzerland; 2Paediatric Infectious Diseases Unit, Paediatric Department, Children’s Hospital Lucerne, Cantonal Hospital Lucerne, 6000 Luzern, Switzerland; 3Faculty of Health Science and Medicine, University Lucerne, 6002 Lucerne, Switzerland; 4Department of Pharmaceutical Sciences, Division of Pharmaceutical Technology, University of Basel, 4056 Basel, Switzerland

**Keywords:** patient safety, child-appropriate formulations, orally fast disintegrating tablets, pediatrics, formulation, excipient, worldwide clinical need

## Abstract

A majority of therapeutics are not available as suitable dosage forms for administration to pediatric patients. The first part of this review provides an overview of clinical and technological challenges and opportunities in the development of child-friendly dosage forms such as taste masking, tablet size, flexibility of dose administration, excipient safety and acceptability. In this context, developmental pharmacology, rapid onset of action in pediatric emergency situations, regulatory and socioeconomic aspects are also reviewed and illustrated with clinical case studies. The second part of this work discusses the example of Orally Dispersible Tablets (ODTs) as a child-friendly drug delivery strategy. Inorganic particulate drug carriers can thereby be used as multifunctional excipients offering a potential solution to address unique medical needs in infants and children while maintaining a favorable excipient safety and acceptability profile in these vulnerable patient populations.

## 1. Introduction

A majority of therapeutics prescribed for the prevention or treatment of diseases in infants and children are medicines designed for and studied in adults. They may not be the most effective, and/or are delivered as non-palatable dosage forms eventually leading to poor patient adherence and inadequate drug exposure. In the 1960s, the acknowledgement of children as “therapeutic orphans” led to a worldwide wake-up call. The need to conduct clinical trials with medicines utilized in infants and children was recognized to be an extremely important way to improve the health of children, especially in areas of high unmet clinical need [1]. In 2007, the World Health Organization (WHO) launched the “make medicines child size” (MMCS) campaign by urging countries to prioritize procurement of medicines with appropriate strengths for children’s age and weight. In addition, it was suggested to develop child-friendly formulations such as multi-particulate oral formulations [2].

According to Kaushal et al., about 7.5 million preventable medication errors occur with pediatric patients in the US each year [3], among which 14–31% result in serious harm or death [4,5]. Medication errors mainly occur in high-risk settings such as the Emergency Unit, Intensive Care Unit (ICU), Anesthesiology and Neonatology Units, where severe diseases as well as life-threatening conditions are being treated with narrow therapeutic index drugs. In infants and children, systemic anti-infectives are the most prescribed and used drugs particularly in the out-patient setting. Within this age population, the <2-year-old infants have the highest drug prescription prevalence (i.e., 2.2–4.7 prescriptions per person per year) [6]. Therapeutic errors frequently occur, following parenteral infusions, oral fluid administration, tablet splitting, tablet crushing, and unlicensed and off-label use of drugs with doses extrapolated from adult literature [7]. Human errors associated with system defects and lack of clinical pharmacists in hospitals are other well identified risk factors for medication errors throughout the whole dispensation chain of drugs (i.e., prescription, transcription, dispensing, dosage, administration, compliance monitoring). Child-friendly orally available formulations are necessary, to meet the goal of increased out-patient treatment. As soon as hospitalized children recover from an acute infection and show adequate ability to drink and eat, they should continue their treatment with an oral agent in the out-patient setting. For antibacterial drugs, an early switch from i.v. to oral treatment is possible and also effective [8].

Although much progress has been made in recent years, the development and utilization of drugs in newborns, infants and older children is still associated with a wide range of pharmacological and clinical challenges. The lack of funding, small market size, ethical issues, specific ethical concerns and uniqueness of children’s physiological, developmental, psychological, and pharmacological characteristics makes conducting clinical trials in pediatrics more challenging than in adults [9]. Less than 50% of drugs entering the market are clinically evaluated in the pediatric age group [10]. As pediatric patients are highly diverse, ranging from preterm, term neonates to adolescents, they differ markedly from adults because of developmental changes and continuous growth affecting pharmacokinetics (absorption, distribution, metabolism, and elimination), and pharmacodynamics (desired and undesired effects). In addition, target values of biological parameters or drug concentrations can differ greatly between adult and pediatric patients, as well as between healthy and sick children. Furthermore, formulations are frequently designed for adults and hardly ever for optimal use in children. These factors make it difficult to design pediatric studies, find optimal pediatric dosing, and select an age-appropriate formulation.

Drug prescription in children is often based on extrapolation from clinical trials in adults. Formulations are hardly ever designed for an optimal use in children. Large capsules and tablets (e.g., 6-mercaptopurine, temozolomide), poor taste, high number of dose units, administration volumes and safety of excipients limit the acceptability of many dosage forms in pediatrics and may have an impact on bioavailability [11]. As such, there is a need to investigate innovative individualized treatment and care as well as child-friendly oral formulations. Furthermore, novel formulations should take into consideration the situation in low- and middle-income countries, namely compatibility with high and/or humid temperatures, inefficient transport systems and interrupted supply chains, as well as poor storage conditions.

An optimal pediatric formulation should meet the following requirements: low frequency of dosing, an appropriate dosage form for various pediatric age groups, convenient and reliable administration, minimal impact on lifestyle and daily routines, use of non-toxic and well tolerated excipients, taste masking, and cost-efficient manufacturing [12]. In summary, introducing child-friendly, age-appropriate, formulations is part of an innovative approach to enhance drug acceptability and to improve drug adherence and clinical outcomes.

The first part of the present review focuses on clinical and technological challenges and opportunities associated with the design of child-friendly formulations. This includes factors contributing to patient acceptance, dosage form design, the choice of excipients, the impact of dosage forms on pharmacokinetics and pharmacodynamics, regulatory aspects, economics and sustainability. Practical implications of these factors are highlighted by examples from the clinical practice in the form of box-inserts. Illustrations taken from our clinical experience or from the literature will thus provide a comprehensive overview of the challenges and hurdles that paediatricians face in their daily clinical practice. Table 1 summarizes these factors.

The second part of the review discusses the example of orally dispersible tablets as an innovative and novel strategy to overcome the above mentioned challenges. In particular, we will introduce porous inorganic drug carriers as a novel multi-functional excipient for pediatric use. Alternative formulation strategies are presented in Table 2, but are not discussed in detail as several excellent review articles have already summarized these established and traditional technologies [13,14].

## 2. Methods

Literature search was conducted on PubMed database between November 2022 and February 2023. Keyworks used in the literature search were: “medication error(s)” or “medication mistake(s)” or “formulation(s)” or “dosage(s)” or “excipient(s)” or “orally dispersible tablet(s)” or “child-friendly” or “drug adherence” or “developmental pharmacology” or “paediatric(s)” or “child” or “neonate(s)” or “infant(s)” or “adolescent(s)”. The reference lists of the selected papers were also reviewed in order to identify additional relevant studies.

## 3. Clinical and Technological Challenges

### 3.1. Palatability and Taste Masking

In pediatric patients, unpleasant, bitter taste of a medicine is one of the most frequent causes of reduced drug-adherence, treatment failure and the importance of building palatability into pediatric medicines is now recognized by the pharmaceutical industry and the regulatory authorities (Table 1). For this reason, medication palatability is a key element of therapeutic drug-adherence and successful therapeutic outcome in pediatrics where the ability to swallow prescribed drugs can influence the choice of a given medicine or formulation by the treating clinician [11] (Box 1, Clinical case study 1).

Box 1Clinical case study 1–Use of midazolam formulations in pediatrics.    Midazolam is one of the most extensively used short-acting benzodiazepine for anxiolysis and pre-sedation in clinical practice. In the Pediatric Anesthesiology or Emergency Units, midazolam is mainly administered as an unlicensed oral syrup or rectal suppository. However, due to its very unpleasant taste, oral midazolam syrup has a rather low acceptability, especially in children who may refuse to swallow it and spitting of the drug is not infrequent in children. Therefore, an additional sweetening agent to mask the bitter taste is often used. In the past, orally dispersible tablets formulation have shown a high benefit in the management of seizures, e.g., sublingual midazolam (Buccolam) with a faster onset of action than oral midazolam syrup is an approved treatment of convulsive seizures in pediatrics. However, it is less cost-effective than conventional oral midazolam syrup and has a similar unpleasant bitter taste, which precludes its successful usage as premedication in children. The use of midazolam rectal suppository leads to a high variability in the onset of sedation and potentially delayed procedures. Medication palatability is a key element of treatment adherence and successful therapy outcome, especially in children where the ability to intake prescribed drugs can have major repercussions on the choice of a given medicine or formulation by the treating clinician.


pharmaceutics-15-01033-t001_Table 1Table 1Challenges and opportunities associated with the development of child-friendly formulations.
ChallengesOpportunitiesClinical Case StudyImpact of growth and development on drug ADMERapid and continuous growth and development in infancy and childhoodGradual organ maturation, at different rates Changes in body surface area and weightPharmacokinetics, pharmacodynamic response to substances and adverse reactions vary with ageDisease may present differently than in adultsNeed for thorough understanding of maturation processes during infancy and childhoodNeed to express dose and dose frequencies as function of age group4Registration of novel pediatric formulationsBioequivalence studies usually conducted in the adult populationRegulations and ethical hurdlesSlow advancement in child-friendly dosage formsImplementation of Best Pharmaceuticals for Children Act (BPCA) and introduction of Pediatric Investigation Plan (PIP)
Excipient safety and acceptabilityElevated toxicity and safety risks for preterm and term newborns and infants < 6 monthsMinimum, non-toxic excipientsThorough assessment of toxicity and risks, acceptable daily intake levels, purity and tolerability while selecting excipients3, 4Palatability and taste maskingUnpleasant taste of a medicine: one of the most frequent causes of treatment failure in pediatric patientsRejection of bitter taste and preference for sweet taste in the pediatric populationInefficiency of sweetenersTaste-maskingRapid disintegration of ODTs, rapid release of API to taste budsComplexationPhysical shielding1, 2Administration flexibilityLack of flexible solid oral dosage formsMinimal dosage frequencyMinimal impact on lifestyleConvenient, easy, reliable administrationODT dosage easily administered to children and elderly patients without the need for hospitalization or the support of medical professionalsODT offer high degree of flexibility (tablet can be pre-dispersed or directly disintegrated within the oral cavity or ingested) Direct access to the systemic circulation bypasses the gastrointestinal tract5Tablet sizeChallenges of swallowing tablets or capsules in extreme agesHigh dosage formulations limiting tablet size minimizationContent uniformityPrecise dosingODT size minimizationDirect compaction for high drug loadingBalance between ODT hardness and disintegration1Onset of action and emergency situationsLong onset of action in emergency situations Unpracticable formulations (syringes)ODT disintegration time < 30 secondsODT potential use in pain, epilepsy, anaphylaxis
Sustainability and economicsLimited access to water, poor hygiene, heat, humidity in LMICsTransportation risksHygroscopic taste masking sugarsPoor handling of drugs leads to development of resistances, hospitalization prolongation and higher need for healthcare resourcesEasily produced, stableCost and commercial viabilityNo need for water with ODT (rapid disintegration with saliva) ODT stabilityODT low bulk and weight5, 6


Taste masking is a well-documented challenge in formulation of oral disintegrating tablets (ODTs) caused by the stimulation of certain receptors in the oral cavity. Several reviews have emphasized the importance of taste masking and palatability in ODTs for ensuring patient acceptance and compliance. Poor adherence to therapy can occur because of a lack of taste masking (Box 2, Clinical case study 2). The rapid disintegration of ODTs in the oral cavity and immediate release leads to early exposure of the active pharmaceutical ingredient (API) to the taste buds, making it difficult to mask the bitter taste commonly associated with many APIs [15,16,17,18]. This challenge is further intensified by differences in taste perception between adult and pediatric patients, which is especially characterized by a rejection of bitter-tasting and a preference for sweet-tasting foods in pediatrics [19]. Consequently, a great effort in formulation development must be invested in utilizing excipients for taste-masking of bitter APIs. A common strategy for taste masking is the use of artificial sweeteners in combination with flavors, as they are well known, widely available and typically do not affect the release of the API. However, sweeteners are not efficient in masking bitterness, requiring use of large amounts or combination with different taste masking strategies. Improvement in taste can be also achieved with chemical interactions by complexation (e.g., cyclodextrines). Physical shielding by coating of the API is considered most effective but can affect bioavailability, is expensive and more technologically challenging as it requires dedicated processes and equipment [16,20].

Box 2Clinical case study 2–Lack of palatability of tuberculosis treatment in toddlers.    Siblings (3 year old female and 2 year old male) were hospitalized for the diagnostic procedures and treatment initiation for pulmonary tuberculosis. They were prescribed isoniazid, rifampicin and pyrazinamid. Isoniazid and pyrazinamid were available as tablets (crushed for administration) and rifampicin as syrup. The children were discharged after 3 days of observed treatment in the hospital. Considerable help from the paediatric nursing staff to assist the mother to administer the medication was required. One day later, the single mother presented with the children to the emergency unit complaining that the children were not taking the medication. At each administration, they had been spitting the crushed tablet and the syrup out. The children were readmitted. Medication swallowing training was installed with the extensive help of nursing staff and paediatric psychologists. After a long hospitalization of 8 weeks, the children were able to tolerate the administration procedure without assistance by nursing staff and psychologists and could be discharged home to continue treatment for a further period of 4 months.

### 3.2. Flexibility of Dose Administration

Flexible solid oral dosage forms that would allow to adjust dose depending on body weight and age groups are considered most suitable for children at the global level especially for developing countries [21]. Flexible solid oral dosage forms include tablets that are dispersible and can be used for preparation of oral liquids suitable for the younger age groups, powders, granules, and pellets [22]. In this context, ODTs have been widely studied and are recognized as a popular and effective dosage form for vulnerable patient populations, including children and the elderly (Table 1). This dosage form can be easily administered to children and elderly patients without the need for hospitalization or the support of medical professionals, and it has been shown to be well-tolerated and safe [23]. Studies have demonstrated that solid dosage forms such as mini tablets or ODTs are more accepted by children than syrup formulations, which are considered the gold standard in pediatric drug delivery [14,24,25]. Furthermore, ODTs offer a high degree of flexibility in terms of administration methods, as the tablet can be pre-dispersed or directly disintegrated within the oral cavity, or even ingested in its whole, depending on the individual’s preference [26]. Oromucosal delivery systems are designed to specifically target the highly vascularized oral mucosa for buccal drug delivery using ODT received a lot of attention in recent years. Direct access to the systemic circulation bypasses the gastrointestinal tract, thus preventing hepatic clearance, such as with ororodispersible desmopressin tablets [27]. As a result, bioavailability increases, allowing lower doses to be administered and reducing side-effects and systemic toxicities (Table 2).


pharmaceutics-15-01033-t002_Table 2Table 2Formulation advantages and disadvantages in the pediatric population.FormulationsPhysiological Differences in Children Compared to AdultsAdvantagesDisadvantagesOralLiquid: solutions, suspensions, syrups, emulsionsTolerance of smaller fluid volumes, delayed onset of absorption and reduction in drug bioavailability [28] Higher permeability of mucosa [29]Maximal dosing flexibility (graduated pipettes and measuring spoons)Volume sizeTaste and palatability issues [30] Hygienic and water availability issues in LMICsSolid: tablets, capsules, powders, granules, pellets, sprinkles, chewable tablets, orodispersible tablets, oral lyophilisates, oral films, controlled release tablets
Enhanced stability compared to liquid formulationsSmaller size tablets more acceptable, suitable for highly soluble drugsTablet, capsule swallowing not tolerated in young children Risk of aspiration or chokingRisk of expulsion/expectoration Delayed onset of action if solid material needs to dissolve prior to absorptionNasalNebulizers, aerosolsSmall nasal cavityObligate nose breathers < 6 months Direct access to systemic circulation Fast onset of action (close to IV) Non-invasive, painless administrationModerate acceptability High variability in exposureIrritation of the mucosa OcularDrops, ointments, gels, insertsAdult eye anatomy and physiology from 3-4 years [31] Infant tear film (volume/protein content) decreased [32] Higher drug absorption and corneal permeation in neonates [33] Non-invasive, painless administrationNovel ocular drug delivery systems [34]Moderate acceptability in infants and toddlersSuboptimal absorption due to anatomical and physiological constraints [35] OticEar drops, spraysExternal auditory canal straighter, narrower, and shorter in infants [36]Non-invasive, painless administrationModerate acceptability in infants and toddlersRectalSuppositories, Creams, ointments, foams, sprays, enemasAdult anatomy from 10 years [37] Higher exposure in preterm infants [38] Rapid absorption Preferred route while oral route is contraindicated or rejected because of acceptability/palatability issuesSuppositories dose adjustmentsRisk of expulsionLow concordance and drug adherenceIrritation of the rectal mucosa ParenteralIV, IM, SC, intra-dermal injection Reduced skeletal muscle blood flow and inefficient muscular contractions in neonates [39] Higher IM absorption in neonates [40,41] Specific attention to electrolyte concentration for neonates (immature renal function) Age-dependent daily fluid and sodium requirementsInstant absorption, reduced time-to-effectHigh concentrations in less perfused tissues Formulation of choice in central nervous system diseases Variety of sizes and concentrations Accurate dose measurementInvasive and painful; needle fear Challenge of vein accessRisk of paravenous drug administration and tissue damage [42] Risk of systemic adverse effects (e.g., osmotic overload)DermalLotions, shampoos, ointments, creams, powders, transdermal patchesImmature stratum corneum <12 months Thin and well perfused skinHigher surface body area to body weight ratio in neonatesLower volume of distribution in children [43] Fever increases permeation rateHigh patient complianceContinuous, painless active drug permeation over hours (transdermal patches)Accidental removal, rubbing, touchingIrritation and subsequent infectionPulmonaryPressurized metered dose inhalers, dry powder inhalers Airway sizeRespiratory rateInspiratory/expiratory flow ratesBreathing patternsLung capacitiesNon-invasive, painless administration Avoidance of hepatic first-pass metabolismAlternative route to parenteral application for systemic treatment with peptides and proteinsModerate acceptability in infants and toddlersInstruction/training for administration by professional neededMinimal inspiratory flow required Variability on the fraction delivered to the lungs 


### 3.3. Excipient Safety and Acceptability

Excipients are used to optimize the formulation, improve palatability (influenced by drug-s crystalline structure and solubility), shelf-life and manufacturing processes. When selecting excipients, it is crucial to conduct a thorough assessment of toxicity and risks, considering regulatory compliance, acceptable daily intake levels, purity, tolerability, and the age of the intended patient population. Even excipients generally recognized as safe (GRAS) can be unsafe in young children due to their immature ADME processes [44]. Developmental pharmacology concerns related to excipients have been raised, with elevated toxicity and safety risks for preterm and term newborns and infants younger than 6 months of age [13,45]. Use of several excipients hold significant safety warnings in pediatrics as is the case for benzyl alcohol used as preservative, which can lead to neurotoxicity and metabolic acidosis. Ethanol may also lead to neurotoxicity and cardiovascular issues; propylene glycol to neurotoxicity (Box 3, Regulatory case study 3), seizures and hyperosmolarity; polysorbate 20 and 80 to liver and kidney failure; sucrose to dental caries and azo dyes to hyperactivity; acetem (acetylated mono- and diglycerides) present in syrup can solve plastics (e.g., when administered by nasogastric tube or PEG connection, PVC connections).

Box 3Regulatory case study 3–FDA black box warning on Kaletra.    Kaletra (lopinavir/ritonavir) oral solution contains the excipients alcohol (42.4% *v*/*v*) and propylene glycol (15.3% *w*/*v*). When administered concomitantly with propylene glycol, ethanol competitively inhibits the metabolism of propylene glycol, which may lead to elevated concentrations of propylene glycol. Preterm neonates may be at increased risk of propylene glycol-associated adverse events due to diminished ability to metabolize propylene glycol, thereby leading to accumulation and potential adverse events. FDA conducted a review of the Adverse Events Reporting System (AERS) database from the approval of Kaletra oral solution in September 2000 through September 2010. The review yielded 10 post-marketing cases with life-threatening events reported in neonates who were predominantly born preterm (8 of 10 neonates were born at gestational ages ranging from 28 to almost 35 weeks) and who received Kaletra oral solution. Post-marketing life-threatening events included cardiac toxicity (including complete AV block, bradycardia, and cardiomyopathy), lactic acidosis, acute renal failure, central nervous system (CNS) depression, and respiratory complications. Of the 10 cases, there was one death due to cardiogenic shock related to a large overdose of Kaletra oral solution. From these cases, it appears that neonates taking Kaletra oral solution, especially those born prematurely, were at risk of lopinavir, ethanol, and/or propylene glycol toxicity [46]. The total amounts of alcohol and propylene glycol from all medications that are to be given to pediatric patients from 14 days to 6 months of age should be taken into account in order to avoid toxicity from these excipients.

### 3.4. Tablet Size

Swallowing of traditional tablets, designed for adults, is often a problem for young children and patients with underlying comorbidities [47,48]. In infants aged 6 to 12 months, 2 mm diameter tablets are considered acceptable [49]. For children aged 2 to 6 years, tablets less than or equal to 3 mm diameter are considered suitable [50]. As such, the size of an ODT must be minimized to ensure safe administration and increase acceptability in pediatric patients. This constraint significantly reduces the potential drug dose. For example, oral administration of drugs requiring high doses, such as many antibiotics, is not feasible in the form of solid dosage forms any longer. In addition, the amount of used excipients becomes a limiting factor. In particular, the formulation of ODTs relies on excipients for taste masking, super-disintegrants, fillers, binders for mechanical strength, and lubricants [51]. These excipients not only affect the tablet size, but also the amount of API that can be incorporated into the formulation. They are necessary to ensure proper performance such as rapid disintegration and taste masking [17,52]. Therefore, ODT formulations are mainly used to formulate highly potent active pharmaceutical ingredients. The major challenges in this regard are content uniformity and precise dosing. Manufacturing of low-dose ODTs necessitates additional processing steps, such as granulation, which can increase time and cost associated with production. In addition, it is critical to use a well-designed and validated blending process and to control drug adhesion to surfaces during manufacturing process [14]. In this respect, direct compaction is a method of choice for the large-scale production of ODT, as it is a continuous, efficient, and cost-effective technique [53,54].

As outlined above, ODT-based drug administration is a particularly convenient solution for infants and young children, who notoriously have problems swallowing tablets or capsules. In particular, orally disintegrating mini tablets (ODMTs) are a promising dosage form for pediatric drug delivery. However, the manufacturing of ODMTs is more demanding as compared to the production of larger tablets. It requires an excellent understanding of the characteristics of excipients such as particle size distribution, flowability, friability, compactability, and wettability [53,55,56]. Ideally, ODTs are characterized by a high porosity, allowing for a rapid uptake of liquid. However, high porosity is often associated with insufficient hardness and high friability. When formulating ODTs, it is therefore necessary to find a compromise between hardness and disintegration behavior [57].

### 3.5. Onset of Action and Emergency Situations

Parenteral administration of drugs by, for example, intravenous injection is the gold standard if a rapid onset of action is required (Table 2). Auvi-Q (epinephrine injection) is a low dose epinephrine auto-injector for the emergency treatment of allergic reactions in infants and toddlers. However, this approach is expensive due to the use of an auto-injector and the product has a limited shelf-life. Traditional oral formulations have a time to reach clinical efficacy, which is often too long in emergency situations, such as resuscitation and intensive care.

ODTs are a type of solid oral dosage form that rapidly disintegrate within the oral cavity, typically within a few seconds. The Food and Drug Administration (FDA) recommends a disintegration time <30 s for a tablet to be classified as ODT [58]. Orally disintegrating tablets are frequently formulated to elicit a more rapid onset of therapeutic action since drugs are taken up by the buccal mucosa. This is crucial in certain acute or emergency conditions, such as pain, tonic-clonic seizures, status epilepticus, and anaphylaxis [47,48,59]. It should be noted that differences in rate of absorption of a given drug affects time (t_max_) to maximal drug concentration (C_max_); e.g., liquids have a more rapid onset of action than tablets (no disintegration step), while differences in the extent of absorption affect the C_max_ or area-under-the-curve (AUC).

### 3.6. Impact of Developmental Pharmacology on Drug Absorption, Distribution, Metabolism and Elimination (ADME)

Pediatric patients are not small adults. Pediatric patients are a heterogeneous population ranging from preterm and term neonates, infants, older children to post-pubertal adolescents [60]. As such pediatric patients differ markedly from adults in the sense that developmental changes and continuous growth have an impact on body composition, organ maturation and physiological and biochemical processes that govern the pharmacokinetics (absorption, distribution, metabolism, and elimination) and pharmacodynamics (desired and undesired effects) of medicines, as well as pharmacogenomics (e.g., gene switching during development or different isoforms from post-translational spicing during development) (Table 2). In neonates, gastric pH approximates neutral pH values directly after delivery and then decreases to acidic pH values shortly after birth [61,62]. Slower gastro-intestinal but faster intramuscular absorption in infancy surely influence the choice of route of administration of a given drug. Body weight and composition also dramatically changes in the first months of life. Drugs distribution depends on the body composition and on the physio-chemical properties of a given drug. For instance, hydrophilic drugs have a larger volume of distribution in newborns due to their higher percentage of extracellular water (around 70 to 80%). Due to limited protein binding in infants, newborns might thus require a higher dose per kilogram of bodyweight to ensure effective distribution through tissue and plasma [63]. A larger brain/body weight ratio and higher blood–brain barrier (BBB) permeability in younger children leads to high drug intake of drug able to cross the BBB. Different metabolic pathways and drug-metabolizing enzymes (DMEs), e.g., cytochrome P450-dependent enzymes, show various and non-uniform maturation profiles during first months of life [64,65]. For instance, the expression of CYP3A7 is high during fetal life and its activity decreases within two years after birth, whereas CYP3A5 becomes more active later in pediatric life [64,66]. A larger liver/body weight ratio in infants and various iso-enzyme specific maturation schemes lead drugs to metabolize differently according to age, e.g., variation in relative degree of glucuronidation and sulphuration of paracetamol (Box 4, Clinical case study 4). The glomerular filtration rate (GFR) is principally accountable for the filtration and elimination of drugs and their metabolites and reaches adult values by the end of the first year of life. Maturation and developmental pharmacology also influence adverse drug reactions, e.g., increased valproate hepatotoxicity in young children, tetracycline stain of developing enamel, chloramphenicol and grey baby/infant syndrome (variable conjugation by UDP-glucuronyltransferase isoforms). On the other hand, less hepatotoxicity is seen in children taking tuberculostatic drugs (isoniazid, rifampicin, pyrazinamide) compared to adults. This is also seen with higher doses, which are necessary to treat tuberculosis effectively in children [67]. Pediatric diseases may be unique to infants and/or older children or may present differently in children than in adults, e.g., genetic epilepsies associated with channelopathies in children show a better response to standard drugs, haematogenous bone and joint infection in children. As such tailored, individualized dosing strategies are even more critical in children than in adults. In bone and joint infections, an early switch to oral antibacterial treatment can be made in children, henceforth, there is the challenge of an optimal oral systemic treatment modality [8].

Box 4Clinical case study 4–Use of ophthalmic briomonidine formulation in pediatrics.    A 1 month-old infant treated with biomonidine eye drops for glaucoma due to Peter’s anomaly (congenital eye disease complicated with opaque cornea, risk for glaucoma, cataract, or retinal detatchment) developed recurrent episodes of unresponsives, hypotension, bradycardia.    Elevated systemic plasma concentrations of brimonidine 1459 pg.mL^−1^ and 700 pg.mL^−1^ were retrieved following ophthalmic instillation, compared with reported adult studies that show a maximum concentration of 60 pg.mL^−1^, leading to somnolence or coma [68].

### 3.7. Novel Formulations and Pediatric Clinical Development

There are many barriers to pediatric drug development, including ethical concerns and economic barriers. Development of child-appropriate formulations can be time-consuming and cost intensive, due to aforementioned challenges and potential need to develop more than one formulation to allow easy administration to pediatric patients across all age groups. In addition, manufacturing cost of specialized formulations, especially when weighed against return on investment and particularly for small markets or niche products, can be high compared to well-established, non-complex formulations manufactured on a large scale. Furthermore, once a new drug received marketing authorization for adults, several years are often needed to obtain a pediatric indication. Indeed, generating enough data for finalization of Paediatric Investigation Plans (PIP) is often delayed.

To encourage development of pediatric-friendly dosage forms, such as ODTs, various regulatory incentives such as the Best Pharmaceuticals for Children Act (BPCA) have been established and the PIP was implemented by health authorities. Ideally, a pediatric-appropriate formulation should be bioequivalent to an adult product to minimize prescription errors and enable switching of formulations at a given age. A review from Batchelor et al. showed that current pediatric formulations were not equivalent to the reference adult product in 40% of cases [69].

Selection of a dosage form and its composition are pivotal aspects of regulatory documentation. However, it should be noted that a decade after initiation of the PIP, increase in availability of child-friendly dosage forms on the market has yet to materialize and significant deficiencies persist [44,70,71]. To date, one official regulatory document, the “Guideline on Pharmaceutical Development of Medicines for Pediatric use” has been published by the European Medicines Agency (EMA) [14].

Lack of suitable delivery system is one of the main reasons for trial failure in pediatric drug development [72]. Drug adherence, palatability and acceptability are even more important factors in pediatric than in adult drug development. A recent survey on drug-handling issues and expectations in parents and children shows that this topic is of high concern and an often-encountered situation. Medication related factors, particularly administration and drug formulation play a key role. The oral route is the preferred method of drug delivery [73]. As such, appropriate taste-masking is essential to overcome these challenges in pediatric studies.

### 3.8. Socioeconomic Aspects

Stability is a formulation property that ensures uniformity in drug administration and optimal drug preparation management in hospitals. For instance, five different hydrochlorothiazide oral formulations prepared with traditional compounding techniques in hospital pharmacies to treat heart failure and edemas in babies were compared and subjected to quality control tests (pH, particle size, viscosity, dose content and stability). Only one studied formulation met the defined quality criteria and allowed for a correct dose to be administered. Shelf-life was 3 weeks when stored at 5 °C and protected from light [70].

In the majority of low- and middle-income countries (LMICs), water is either lacking or a limited resource. Drug reconstitution is therefore often problematic. Rapid disintegration of ODTs upon contact with saliva eliminates the need for drinking water, making them an attractive dosage form for pediatric patients, particularly in developing countries [74]. Poor hygiene and environmental limitations (heat, humidity) are additional constraints in drug administration (Box 5, Clinical case study 5). Liquid dosage forms are not desirable for the hot and humid tropical conditions found in most LMICs. Furthermore, they require bulkier packaging than solid formulations. Fragmented transportation systems found in many LMIC countries prevent distribution of refrigerated products. In contrast to liquid dosage forms, ODTs are solid dosage forms associated with improved chemical stability. This facilitates storage and logistics in harsh climate conditions [23].

The WHO has stated that solid dosage forms are the preferred treatment option for children. This includes tablets that are oro-dispersible or solid dosage forms, which can be used for the preparation of orally administered suspensions or solutions. This is in particular recommended for children in developing countries.

The incorporation of hygroscopic excipients, such as sweeteners used for taste masking, in combination with high tablet porosity is problematic. This can lead to stability issues since the tablets can absorb significant amounts of moisture. This results in compromised functionality or reduced structural integrity [15,16,18,47,75]. Lyophilisates, such as the Zydis^®^ ODT technology, are particularly susceptible to humidity. The problem can be alleviated by the use of specialized and costly primary packaging [44,57].

Box 5Clinical case study 5–Use of ivermectin formulations in children.    A family (3 children: 2, 4 and 7 years and parents) presented to the emergency department with pruritic skin lesions. The physician suspected and confirmed the diagnosis of scabies. In Switzerland only topical anti-scabies treatment modalities are available. Systemic ivermectin is not registered. He prescribed topical permethrin cream 5% instructing the parents to administer this for every family member and including every body part (head-to-toe) and leaving the topical treatment on the skin for at least 8 h. This procedure should be repeated in 7 to 10 days. The family returned to the ER after three weeks and complained of continuous itching and persistant skin lesions. They reported that it was difficult to administer the cream to every body part. One of the children was reluctant and fighting the application. Another wanted the cream to be washed off after 2 h. In the meantime friends of the family were now also complaining of skin lesions and itch.    Currently, in young children, only permethrin is indicated for the treatment of scabies. A child-friendly oral formulation would be a more suitable treatment modality as illustrated in the case. In LMICs, ivermectin may be administered as suspension, or tablets. However, ivermectin in suspension is not practicable as the stability is fragile, the shelf-life is very short, and the suspension is affected by UV light exposure [53]. In addition, tablets offered to young children as crushed or in a suspended form are prone to imprecise dosing (loss of product after crushing or sedimentation of product after suspension). They are not palatable, and thereby prone to be expelled out of the mouth by the child.

The ivermectin example demonstrates that child-friendly dosage forms provide a convenient option for oral drug administration and are expected to enhance drug-adherence in pediatric patients. This example shows as well that tailored formulations for children do not exist in many situations. Clinicians are therefore forced to prescribe and administer adult dosage forms in an off-label manner, by manipulating existing adult formulations such as crushing of tablets designed for adults (Box 6, Clinical case study 6), opening capsules, or applying injectable solutions by other routes. Still, such handling of drugs may have an irrevocable impact on pharmacokinetics, pharmacodynamics, and or safety of a medicinal product [60]. Pediatric infectious diseases formulations not designed for children can promote development of resistances or prolongate clearing of the infection. This results in longer hospitalization, additional interventions, the use of healthcare resources (nursing staff), and higher healthcare costs.

Box 6Clinical case study 6–Use of crushed valganciclovir tablets in pediatrics.    A 4 week old infant was diagnosed with symptomatic congenital cytomegalovirus (CMV) infection and oral valganciclovir was prescribed as per recommendation for a period of 6 months. In Switzerland, only tablets and capsule formulations are registered. Bioavailability of these dosage forms variable and cannot be compared. This is problematic in view of potential carcinogenicity and teratogenicity necessitating a precise dosing. However, the treatment had to be started. Tablets had to be crushed by healthcare professionals and parents and brought into solution for the infant to swallow. The treatment had to be started however. It took 1 month to order and import the oral solution.

## 4. Novel Multifunctional Excipients for the Design of Orally Dispersible Tablets

### 4.1. Excipient Design for Orally Dispersible Tablets (ODT)

Utilization of conventional excipients in the development of ODTs has been found to be inadequate to meet current clinical needs and requirements by health authorities. New excipients are necessary for successful formulation of ODTs. Efforts in developing new excipients can be classified into three categories: modified, co-processed, and novel excipients. The development path for each class of excipient varies. For instance, development of novel excipients is a costly and time-consuming process, typically lasting 6 to 7 years requiring compliance with regulatory and safety standards. Due to these factors, development of co-processed excipients with pre-approved functions is often preferred by pharmaceutical industries, as it simplifies the regulatory approval process, reduces development costs, and increases likelihood of success [76].

Co-processed mannitol is a commonly utilized excipient for the direct compaction of ODTs due to its slight sweetness, taste-masking properties, non-hygroscopic nature, and improved disintegration characteristics in comparison to polyoles. Despite the availability of pre-formulated mixes such as Pearlitol^®^ Flash, Ludiflash^®^, Parteck^®^ ODT, Pharmburst^®^ 500, and Prosolv^®^ ODT, some of the mentioned ready mixes reveal a poor disintegration behavior even at very low compressive forces. Their use as excipients for ODTs is therefore limited [55]. Alternative pre-formulated mixes for direct compaction, such as Orasolv^®^, which uses effervescent-based disintegration, exhibit low mechanical strength, are susceptible to moisture, and necessitate specialized packaging. Subsequent development has led to Durasol^®^, which demonstrates improved compactibility due to its composition of primarily nondirect compression fillers, including mannitol, lactose, sorbitol, and sucrose. However, the primary drawback of this technology is its limited capacity for drug incorporation. Its use is therefore limited to the formulation of small doses only [77].

### 4.2. Multifunctional Porous Calcium Carbonate/Phosphate Carriers

The preceding section emphasized technological difficulties associated with formulation of ODTs and that current excipients are inadequate in providing the necessary performance, even when co-processed. Consequently, there is an urgent need to create novel excipients that possess combined actions. In the development of a new excipient, it is essential to consider needs and perspectives of key stakeholders, including patients, pharmacists, treating clinicians, and manufacturers, to ensure its success. The main challenge lies in the creation of multifunctional excipients that can produce hard, rapidly disintegrating tablets that are easily processable, safe, and do not add to the already burdensome regulatory filing process [78].

One particularly promising multifunctional excipient that gained significant attention is functionalized calcium carbonate (FCC) [79]. Inorganic porous carriers are a promising technology due to their stability, well-defined surface properties, high pore volume, narrow pore diameter distribution and large surface area [80]. These an inorganic porous microparticles have an average size of 10–20 µm and a specific surface area of 30–70 m^2^/g [81]. Unlike mesoporous silica particles, which have been frequently cited for oral drug delivery, FCC is biodegradable and considered safe, as it is a mixture of calcium carbonate and hydroxyapatite, both of which are monographed [78]. The lamellar surface of FCC provides large contact surface, resulting in excellent compressibility and leading to extremely strong tablets and the ability to be dry granulated using a roller compaction. In contrast to traditional excipients, tablets made with porous inorganic carriers exhibit exceptional mechanical strength while maintaining a high degree of porosity. High particle porosity was found to cause rapid disintegration of tablets, surpassing the performance of many commercially available excipients [79,82]. Utilization of an excipient with multiple functionalities can lead to a reduction in the need for additional excipients, thereby improving drug adherence and compatibility in pediatric populations [83,84]. These characteristics demonstrate the potential of porous carriers for pediatric formulations and provide the basis for further clinical evaluation of the excipient’s mouthfeel properties and acceptability. However, as outlined below, the loading capacity of these carriers is limited.

Subjects included in a clinical trial have reported that the disintegrated inorganic carrier-based tablets had a very pleasant mouthfeel, were palatable and highly accepted [84]. In a follow-up cross-sectional acceptability study in 2- to 10-year-old pediatric patients, it was demonstrated that FCC-based ODTs was established as safe, palatable, and highly acceptable in children [46]. The results showed a high acceptability in 93% of the children suggesting that it is well-suited as an excipient for child-friendly drug delivery. Medical personal reported that ODTs were easy and safe to administer [11]. None of the children in this study showed any sign of distress on receipt of such FCC-based ODT.

Given favorable acceptability of the excipient in children, it is important to evaluate stability of the excipient to ensure consistent performance of the formulation and safety. Stability studies of caffeine and oxantel pamoate tablets containing FCC as an excipient demonstrated that they could withstand stress conditions and maintain the stability of the drug substance stable for prolonged periods. However, it was shown that humidity and temperature affected disintegration time, dissolution rate and hardness, emphasizing the need to store tablets at room temperature and in dry conditions [85].

A key feature of porous inorganic calcium phosphate is its ability to encapsulate small molecules, oils, and proteins. Loading can protect sensitive APIs and improve their stability, as demonstrated by loading proteins into FCC which were reported to be stable and preserved their functionalities [86]. Additionally, for poorly soluble molecules, it has been observed that drug loading of porous calcium phosphate results in an improvement of the dissolution rate, attributed to the enlarged surface area and micronization of the drug substance by loading [87,88]. Furthermore, loading of bitter drugs into the pores can also lead to taste masking, which is critical for the formulation of ODTs [89]. The previous discussion has highlighted that one of the obstacles encountered in the development of low-dose ODT containing highly potent API, is the issue of content uniformity. Utilization of porous carriers as a formulation strategy is a suitable approach for addressing this issue as they are typically loaded with API solutions, resulting in high content uniformity [90].

From a drug product formulation perspective, use of inorganic porous carriers as a means of drug loading offers the potential to prevent alterations in excipients performance as the API is encapsulated in the carrier. Characteristics of the tablet are still determined by the excipient and not by the API, as it is encapsulated within the particle [91].This results in the elimination of variability in performance of ODT due to the use of different mono-functional excipients and quantities and allows for more predictable ODT performance and simplified formulation design [54].

A technical limitation associated with FCC is the limited drug loading capacity. During the loading process, it is frequently observed that drug substance depositions occur on the surface of the carrier, leading to modifications in the surface properties. Consequently, FCC loses its multifunctionality, in particular its ability to form hard, rapidly disintegrating tablets [89].

### 4.3. Outlook on Recent Technological Advances in Excipient Design

There is a requirement for accelerating the advancement of innovative excipients that exhibit synergistic properties and possess enhanced functionalities that are appropriate for pediatric pharmaceutical formulations. We firmly believe that porous inorganic carriers are the future for patient-friendly drug delivery and a promising platform technology for the formulation of ODTs [79]. Yet, limited drug loading capacity and potential toxicity of silica-based particles damper the clinical relevancy of these promising strategies. Ongoing research in our laboratory is therefore dedicated to the design of novel types of porous drug carriers by employing a particle-centered approach. Our efforts involve engineering the particle’s geometry to improve the carrier’s drug loading capacity while simultaneously introducing a taste masking functionality. The use of exclusively compendial, biodegradable and biocompatible materials such as calcium phosphate will facilitate the regulatory process and mitigate toxicological concerns. The use of well-established compendial materials is a prerequisite for a future use in vulnerable patient populations. We anticipate that the use of this new generation of excipients will lead to a significant transformation and paradigm-shift in the development of child-friendly dosage forms.

## *5.* Discussion

A significant number of current drug formulations are not suitable for children of different age groups because of heterogeneity of the pediatric population, immature ADME processes, rapid developmental changes, palatability issues, low drug adherence and ethical concerns (Table 3).

Improvements in flexibility of dose administration, tablet size, taste masking, bioavailability, excipient safety and acceptability, stability, manufacturing and affordability represent multiple opportunities at various scales of child-friendly formulations design to overcome clinical and technological challenges in pediatric drug development. Useful findings of this review are summarized in Table 4. Orally Dispersible Tablets (ODTs) are a promising child-friendly drug delivery strategy, offering a potential solution to address unique medical needs in infants and children while maintaining a favorable excipient safety and acceptability profile in these vulnerable patient populations.

## 6. Conclusions

Infants and children require tailored, age-appropriate dosage forms. A new generation of ODT formulations based on inorganic particulate drug carriers may offer advantages of small size, acceptable taste, rapid intra-oral dissolution and absorption with potential for earlier onset of action. Such child-friendly dosage forms will increase drug acceptability and adherence, improve clinical outcomes in infants and children reducing emotional burden in pediatric patients and their families.

## Figures and Tables

**Table 3 pharmaceutics-15-01033-t003:** Factors affecting pediatric drug development.

	Specificities to Pediatric Population to Be Accounted for in Drug Development
Heterogeneous population	Preterm, term neonates, infants, older children, post-pubertal adolescents
Immature ADME processes	Elevated toxicity and safety risks for newborn and infants
Rapid developmental changes	Impact on pharmacokinetics and pharmacodynamics of medicines
Palatability	Rejection of bitter taste
Drug adherence	Difficulty in swallowing tablets
Ethical concerns	Obstacles to include children in research

**Table 4 pharmaceutics-15-01033-t004:** Clinical and technological challenges inherent to pediatric drug development, solid oral dosage forms requirements and orally dispersible tablets (ODTs) as a promising solution to overcome these challenges.

Clinical and Technological Challenges	Solid Oral Dosage Forms Requirements	ODTs as a Promising Solution
Appropriate dosage form	Dose flexibility Uniformity and precise dosing Size and volume acceptability	Minimal size (ODMTs) High porosity
Preparation and administration	Easy handling and reconstitution	Pre-dispersion or direct disintegration within the oral cavity
Drug adherence	Acceptable taste Minimal impact on lifestyle Minimal frequency of administration	Taste masking: use of artificial sweeteners and flavors, complexation
Efficacy and safety	Optimal bioavailability	Rapid disintegration Oromucosal delivery systems
Excipients	Optimal tolerability	Novel excipients Optimal particle size distribution, flowability, friability, compactability, and wettability
Stability	Optimal shelf-life	Improved chemical stability
Manufacturability	Robust process	Direct compaction
Affordability	Acceptable cost to patients and payers Easy storage and transportability	No need for water Optimal packaging, storage and logistics

## Data Availability

Not applicable.

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
