# Peer review of "Meeting Challenges of Pediatric Drug Delivery: The Potential of Orally Fast Disintegrating Tablets for Infants and Children"

_pharmaceutics, 2023, doi:10.3390/pharmaceutics15041033_

Round 1

Reviewer 1 Report

elaborate more on the children perspective, i.e. with reference to their compliance to clinical trials

correct formulations at line 418

improve the conclusions

Reviewer 2 Report

This manuscript describes the pediatric drug delivery, especially orally fast disintegrating tablets foe infants and children. This review manuscript provides comprehensive information on child-friendly formulation at this point. Reviewer thinks the manuscript is to be overall well written. Not found any critical comments.

 I think this manuscript is acceptable for publication by addressing the following minor points.

 Please correct the numbering of case studies so that it starts at 1.

Reviewer 3 Report

The review article on meeting the challenges of pediatric drug delivery is interesting.

I recommend that the authors include a section on barriers to pediatric drug development, emphasizing ethical concerns, economic barriers, and industry perspectives on barriers to pediatric drug development.

I suggest making the article more reader-friendly if they can summarize somewhere why a significant number of current drug formulations are not suitable for children of different age groups.

The inclusion of a clinical case study is an excellent step.

I suggest that authors include a simple table where they can list out the clinical and technological challenges, current practices to overcome them, and then the recommendations in the last column.

A discussion section is missing in the article where they can summarize the details given in several sections.

Also, it would be good to include a section on methods, where they can explain the methods applied for extracting the resources used in this article.
